# The Effects of Age and Body Fat Content on Post-Downhill Run Recovery Following Whole Body Cryotherapy

**DOI:** 10.3390/ijerph18062906

**Published:** 2021-03-12

**Authors:** Adnan Haq, William Ribbans, Anthony W. Baross

**Affiliations:** 1Sports Studies, Moulton College, West Street, Moulton NN3 7RR, UK; 2Sport and Exercise Science, University of Northampton Waterside Campus, Northampton NN1 5PH, UK; billribbans@billribbans.com (W.R.); Anthony.Baross@northampton.ac.uk (A.W.B.)

**Keywords:** whole body cryostimulation, muscle damage, sport, exercise, eccentric

## Abstract

This study explored the effects of age and body fat content on responses to whole body cryotherapy (WBC) following a downhill running bout. Forty-one male participants (mean ± SD age 42.0 ± 13.7 years, body mass 75.2 ± 10.8 kg) were allocated into WBC (n = 26) and control (CON, n = 15) groups. WBC participants were divided into old (OLD, ≥45 years, n = 10) and young (YNG, <40 years, n = 13), as well as high fat (HFAT, ≥20%, n = 10) and low fat (LFAT ≤ 15%, n = 8) groups. Participants completed a 30 min downhill run (15% gradient) at 60% VO_2_ max. The WBC group underwent cryotherapy (3 min, −120 °C) 1 h post-run and CON participants passively recovered in a controlled environment (20 °C). Maximal isometric leg muscle torque was assessed pre and 24 h post-run. Blood creatine kinase (CK) and muscle soreness were assessed pre, post, one hour and 24 h post-run. Muscle torque significantly decreased in both groups post-downhill run (WBC: 220.6 ± 61.4 Nm vs. 208.3 ± 67.6 Nm, *p* = 0.02; CON: 239.7 ± 51.1 Nm vs. 212.1 ± 46.3 Nm, *p* = 0.00). The mean decrease in WBC was significantly less than in CON (*p* = 0.04). Soreness and CK increased 24 h post for WBC and CON (*p* < 0.01) with no difference between groups. Muscle torque significantly decreased in OLD participants (*p* = 0.04) but not in YNG (*p* = 0.55). There were no differences between HFAT and LFAT (all *p* values > 0.05). WBC may attenuate muscle damage and benefit muscle strength recovery following eccentrically biased exercises, particularly for young males.

## 1. Introduction

Whole body cryotherapy (WBC) is an extremely cold treatment (typically below −100 °C) that has been used for pain remission, musculoskeletal disorders and skin lesions [1,2]. Its recent emergence in sport and exercise has added a unique perspective to sports recovery practice [3,4]. Several reported effects of WBC post-exercise include attenuated blood markers such as creatine kinase (CK) [5], muscle soreness [6], inflammation [7,8] and alleviated reductions in muscle torque [9,10]. Despite these effects, the precise impact of WBC in sports and performance remains equivocal [11,12], whilst negative effects have also been reported [13]. Consequently, there remains a need to address the efficacy of WBC in greater depth to better inform the sporting community of its overall merit. 

Inter-individual variability—specifically age and body fat—is a potentially important consideration when evaluating the impact of WBC and informing optimal practices. Probable causes of reduced recovery and performance capacity with ageing include muscle mass and strength loss (sarcopenia), oxidative damage and chronic inflammation [14], which could affect responses to cryotherapy post-exercise. There is a paucity of literature concerning differences in responses to cryotherapy between young and old populations. Cutaneous vasoconstrictor and cold-induced vasodilation following extreme cold can be blunted in older compared to younger individuals [15]. Reduced blood vessel responsiveness and distensibility [16] may be significant in the context of cryotherapy because of the impact on blood redistribution and heat transfer. Therefore, it is conceivable that older individuals would respond to the treatment less optimally than their younger counterparts.

Outcomes to cryotherapy treatments among differing body compositions have not been investigated extensively despite body fat content influencing cold exposure reactions [17]. Higher fat individuals may retain core and tissue temperatures to larger extents following cryotherapy (owing to reduced vasodilation) compared to leaner individuals [18]. Additionally, subcutaneous fat provides thermal insulation and decreases thermal conductance [15]. Thus, the response and overall tolerance to cold temperatures can vary accordingly. A link has been demonstrated between body fat content and duration required to reduce intramuscular temperatures following cold treatments [19]. The theory that higher adiposity could affect the response to WBC is supported by observations of strong negative correlations between body fat percentage and skin temperatures [20,21]. Despite the reported differences in outcomes too cold between different body compositions, the implications of such variances for WBC application post-exercise remains under-investigated. 

This study aimed to examine the effect of WBC following a downhill run, a common exercise protocol imposing continual eccentric contractions on the quadriceps muscles. Such bouts cause muscle torque losses, elevated soreness, plasma CK, inflammatory cytokines and reduced running economy [22,23,24,25], established characteristics of exercise-induced muscle damage (EIMD). One theme of interest is how cryotherapy could potentially alleviate this muscle breakdown and damage, which would present an advantage for athletes since EIMD has negative consequences on locomotor biomechanics and subsequent performance [26,27]. Since downhill running is a whole-body exercise that stresses several physiological systems, it is of interest to determine if WBC could enhance recovery following this modality. Other commonly adopted damage protocols, such as isolated eccentric leg extensions [28] and arm curls [29,30] may be less sports specific, so any WBC-induced responses may be less applicable for general sports recovery. Despite the extensive literature on downhill running and WBC, no study has yet to assess the impact of WBC on recovery following a downhill running bout. Additionally, previous studies that have demonstrated positive effects for WBC treating EIMD (e.g., [6,10]) typically used multiple treatments post-exercise, which are less economical and practical than a single treatment. 

Therefore, the principal aims of this study were as follows:To assess the overall impact of a single WBC treatment on recovery following a downhill run.To assess the impact of age and body fat content on recovery response to a single WBC treatment post-exercise.

It was hypothesised that (1) WBC would attenuate muscle damage markers post-downhill running, thereby supporting its use as a means to enhance recovery after muscle damaging exercise; and (2) younger men with lower body fat contents would respond more positively to WBC than other populations. 

## 2. Materials and Methods

### 2.1. Participants

A sample size calculation (G*Power: significance level 0.05, power 0.8, effect size 0.5) revealed that 9 participants per group would be appropriate to detect an effect. Forty-one male volunteers (mean ± SD age 42.0 ± 13.7 years, height 1.76 ± 0.08 m, body mass 75.2 ± 10.8 kg, body fat 19.2 ± 4.5%) were recruited for the study, which adopted an independent groups design. Participants were randomly assigned as cryotherapy (WBC, n = 26) and control (CON, n = 15). To assess the influence of differing ages and body fat contents, the WBC group was sub-divided into old (OLD, ≥45, mean ± SD age 58.1 ± 7.9 years, n = 10) and young (YNG, <40, mean ± SD age 29.2 ± 7.6 years, n = 13) [31], as well as high fat (HFAT, ≥20%, mean ± SD body fat 23.0 ± 2.9%, n = 10) and low fat (LFAT, ≤15%, mean ± SD 13.8 ± 1.4%, n = 8) groups [30]. Three WBC participants were aged 40–44 and 8 WBC participants had 15.5–19.5% body fat, which were not part of these sub-groups. They were still included in the overall analysis between WBC and CON.

All participants were of a suitable fitness level for the demands of the study, consistently partaking in physical activity a minimum of twice a week. Prior to further screening and assessment, all participants’ blood pressure was assessed and written informed consent was provided. Ethical approval was obtained from the University of Northampton Graduate School Research Ethics Committee and the study was conducted according to guidelines of the Declaration of Helsinki.

Sample characteristics for each group are summarised in Table 1.

### 2.2. Initial Trial

Participants were asked to refrain from alcohol and strenuous exercise for 24 and 48 h respectively prior to all trials. Initially, participants’ anthropometric characteristics were assessed, including height and body mass. Body fat content was assessed by skinfold calipers (Harpenden Indicators, UK) according to the International Society for the Advancement of Kinanthropometry. Four skinfold sites were used: Biceps, triceps, subscapular and iliac crest, with the total skinfold thickness converted into body fat percentage [32]. Participants were familiarised to a muscle torque assessment using a dynamometer (Section 2.5), which involved two submaximal isometric contractions (60% and 80% effort), followed by a singular maximal contraction. 

Maximal aerobic capacity (VO_2_ max) was measured using an online breath by breath analyser (Cortex Metalyser, Germany) calibrated prior to assessment, following an incremental treadmill (Cosmos, Germany) protocol. From a starting speed of 6 km/h and gradient of 1%, the speed increased every two minutes by 2 km/h until 16 km/h was reached. Thereafter, gradient was increased by 2% each stage. Rating of perceived exertion (RPE) [33] and heart rate (HR) were recorded at the end of each stage. The assessment was terminated once the participant experienced volitional exhaustion. The absolute and relative VO_2_ max values were reported and 60% of the absolute VO_2_ max was calculated. 

Participants then completed the muscle torque assessment involving four maximal contractions with the highest torque determined as the pre-torque score. 

### 2.3. Second Trial

Within 3 to 14 days of the first trial, participants returned to the laboratory (ambient temperature 20 ± 0.5 °C) to perform their main trial. Initially, skin temperatures were assessed using four sites—chest (c), triceps/posterior upper arm (a), anterior thigh (t) and calf (ca)—using a thermal imaging camera (FLIR systems, Sweden). Images were captured whilst the camera was held 3.0 metres from the participant and set at an emissivity factor of 0.98. The images were stored using FLIR Tools+ software and analysed based on pre-established regions of interest on the four body regions. The mean skin temperature was calculated as follows [34]:Ts = 0.3×(c+a)+0.2×(t+ca) 

Tympanic temperature was assessed by a thermometer (Braun Thermoscan 7) inserted into the right ear canal. Participants provided a 30µl fingerstick blood sample for the measurement of CK levels using a test strip inserted into a Reflotron Plus analyser (Oberoi Consulting, Derby). 

Muscle soreness and mental wellbeing were assessed via visual analogue scales (VAS) [35]. For soreness, participants squatted against a wall with knees flexed to a 90º angle [35], holding the position for three seconds. Participants marked on the scale how much pain they felt in their upper legs from “no pain” to “pain as bad as it could possibly be”. Mental wellbeing was marked on a scale from “I do not feel comfortable, healthy and satisfied” to “I feel extremely comfortable, healthy and satisfied”. The distances marked were converted to percentages. 

Following a two minute warm up at 5 km/h, participants commenced their 30 min treadmill (HPCosmos, Germany) downhill run at a 15% decline gradient with HR and RPE monitored every 5 min. The treadmill speed was gradually adjusted so that the participants’ average HR throughout the 30 min was roughly equivalent to their target HR. The predetermined HR was extrapolated from the VO_2_ max vs. HR relationship, so that a running intensity corresponding to 60% of their absolute VO_2_ max was maintained. After completing the run, tympanic and skin temperatures were immediately assessed, followed by CK, muscle soreness and mental wellbeing. 

Cryotherapy participants were transported to a therapy centre for their WBC treatment at −120 °C scheduled for 60 min post-downhill run. Control participants remained seated in the laboratory under controlled conditions (20 °C). Thermal images were again taken 5 min post-WBC. Tympanic temperature, CK, muscle soreness and mental wellbeing were measured 10 min post-WBC. Measures for the control participants were recorded at the same time points as stated above. 

### 2.4. Third Trial

Participants returned to the laboratory 24 h post-downhill run. Blood CK, muscle soreness and wellbeing were measured prior to the muscle torque assessment. The study protocol is summarised in Figure 1. 

### 2.5. Assessment of Muscle Torque

Unilateral isometric maximal torque of the right quadriceps was assessed by an isokinetic dynamometer (Biodex Medical Systems 3, New York, NY, USA) calibrated prior to the study. The dynamometer was fitted with a lever arm attachment with a shin pad locked in at an angle of 90º leg extension. Participants sat on the dynamometer chair with 90° hip flexion. The chair was adjusted so that the pivot of the lever arm was located adjacent to the lateral epicondyle of the right leg. The right leg was strapped to the lever arm attachment ensuring the bottom of the shin pad was located just superior to the medial malleolus. The shoulders, trunk and right thigh were strapped tight to avoid excessive body movements [9]. Participants were asked to place their arms across their chests without holding the shoulder straps.

Participants performed two warm up contractions at 60% and 80% effort respectively, (separated by a 30 s recovery period) by exerting force against the pad with their right leg. Following two minutes rest, they performed four maximal contractions (with two minute recoveries) with verbal encouragement given throughout [36]. All contractions were 5 s in duration. The peak torque (Nm) was determined as the maximum of the four contractions. A pilot study conducted in the laboratory revealed a day to day variance of 5.3% within individuals.

### 2.6. Whole Body Cryotherapy Treatment

Cryotherapy treatments were undertaken in a two-stage cryogenic chamber (JUKA, Warszawa, Poland). The source of cold was liquid cryogenic gas originating from external pressure vessels. Participants were screened for contraindications following the completion of a health questionnaire, including hypertension, other cardiovascular diseases, open wounds, cold intolerance, neural/mental disorders and cancers. Before entering the chamber, participants wore a head band, face mask, gloves, socks, elbow and knee bands, and clogs. Verbal instructions were provided. Participants entered the cryotherapy chamber, initially exposed to a vestibule chamber at −60 °C for 30 s, followed by the main chamber at −120 °C for 150 s. On completion, the exit door for the main chamber opened and the participant exited. Thereafter, participants were advised to stay mobile before changing in usual clothing.

### 2.7. Statistical Analyses

All data was analysed using IBM SPSS Version 26. Data for all variables was assessed for normal distribution by the Shapiro-Wilk test. With the exception of tympanic and skin temperatures, all variables significantly deviated from normality and were therefore log or square root transformed as appropriate. A two-way repeated measures ANOVA was used to assess the interaction effect between group (WBC vs. CON; OLD vs. YNG; HFAT vs. LFAT) and time for all major dependent variables: Muscle torque was assessed with a 2 (group) × 2 (time) interaction; soreness, CK and wellbeing were assessed with a 2 × 4; tympanic and skin temperatures were assessed with a 2 × 3. Paired t tests and pairwise comparisons with a Bonferroni correction were applied where necessary to examine differences between specific timepoints. Effect sizes (Cohen’s d) and 95% confidence intervals (CI) were calculated for muscle torque, the main dependent variable of interest.

## 3. Results

Original data (pre-transformation) are presented in figures.

### 3.1. WBC vs. CON

#### 3.1.1. Muscle Torque

There was no significant difference in pre-muscle torque between WBC and CON groups (*p* = 0.27). There was a significant decrease in maximal isometric muscle torque for both groups following the downhill run (WBC, 220.6 ± 61.4 Nm, 95% CI [197.0, 244.2] vs. 208.3 ± 67.6 Nm, 95% CI [182.4, 234.3], *p* = 0.02, d = 0.19; CON, 239.7 ± 51.1 Nm, 95% CI [213.8, 265.5] vs. 212.1 ± 46.3 Nm, 95% CI [188.7, 235.6], *p* < 0.01, d = 0.57, Figure 2). The mean decreases were 12.2 ± 24.8 Nm (6.4%) and 27.5 ± 14.6 Nm (11.5%) for WBC and CON respectively with a significant difference between groups (*p* = 0.04, d = 0.67). The overall difference between groups over time was non-significant (interaction effect, *p* = 0.10).

#### 3.1.2. Muscle Soreness

Soreness significantly increased from baseline to post-downhill run, one hour and 24 h post-run for both WBC and CON groups, (overall effect of time *p* < 0.01 for both groups) with a peak reached at 24 h (47% for WBC; 44% for CON, Figure 3). There was no difference between groups over time (interaction effect, *p* = 0.87).

#### 3.1.3. Creatine Kinase

Blood CK significantly increased from baseline to 24 h post-run for both WBC (157.3 ± 110.4 UI/L vs. 418.4 ± 325.4 UI/L, *p* < 0.01) and CON (176.3 ± 147.0 vs. 553.6 ± 286.1 UI/L, *p* = 0.02, Figure 4). There was no overall difference between groups over time (interaction effect, *p* = 0.78). The mean CK increases (baseline to 24 h post) were 179.7% and 291.4% for WBC and CON participants respectively with no difference between groups (*p* = 0.42).

#### 3.1.4. Tympanic Temperature

There was no difference in tympanic temperature from baseline to post-downhill run in the WBC group. For the WBC group, tympanic temperature significantly decreased post-WBC (36.8 ± 0.5 °C vs. 36.4 ± 0.4 °C; *p* < 0.01). There were no differences for the CON group. There was a significant difference between WBC and CON groups at 1 h post-run (36.4 ± 0.4 °C for WBC; 36.7 ± 0.3 °C for CON, *p* = 0.01, Figure 5).

#### 3.1.5. Skin Temperature

Skin temperature significantly decreased five minutes post-cryotherapy for the WBC group (32.8 ± 0.9 °C vs. 27.3 ± 1.5 °C; *p* < 0.01) whilst there was no difference for the CON group (Figure 6).

#### 3.1.6. VAS Wellbeing

Wellbeing scores did not significantly change between any paired time point for the WBC group, although the overall time effect approached significance (*p* = 0.06). There was no difference for the CON group (*p* = 0.44) and no interaction between group and time (*p* = 0.53). VAS wellbeing scores are displayed in Table 2.

### 3.2. Effect of Age and Body Fat

#### 3.2.1. Muscle Torque

There was a significant difference between age groups for pre-muscle torque (*p* < 0.01). The pre-post difference in torque was significantly affected by age (interaction effect, *p* = 0.02). There was a significant decrease in OLD participants (178.3 ± 37.5 Nm, 95% CI [155.1, 201.5] vs. 155.7 ± 49.2, 95% CI [125.2, 186.2], *p* = 0.04, d = 0.52) but no decrease in YNG participants (257.3 ± 60.7 Nm, 95% CI [224.3, 290.3] vs. 253.3 ± 54.0 Nm, 95% CI [223.9, 282.6], *p* = 0.55, d = 0.07), following WBC (Figure 7A). The pre-post difference in torque was not significantly affected by body fat (interaction effect, *p* = 0.41). There was a trend for a slight decrease in torque for the HFAT group (187.9 ± 31.5 Nm, 95% CI [168.4, 207.4] vs. 167.6 ± 46.6 Nm, 95% CI [138.7, 196.5], *p* = 0.07, d = 0.52) and no decrease for LFAT (247.5 ± 68.0 Nm, 95% CI [200.4, 294.6] vs. 238.0 ± 70.8 Nm, 95% CI [189.0, 287.0], *p* = 0.2, d = 0.14, Figure 7B).

#### 3.2.2. Other Variables

The results for all other variables regarding age and body fat groups are summarised in Table 3 and Table 4 respectively.

There was no overall effect of age (*p* = 0.68) or body fat (*p* = 0.78) on the muscle soreness response. The CK response was not affected by age group (*p* = 0.22) and there was no significant difference between OLD and YNG at 24 h post, when the highest CK value occurred (*p* = 0.13). There was no effect of body fat on CK (*p* = 0.59), including at 24 h post (*p* = 0.16). Tympanic temperature was significantly higher for YNG than OLD at post-WBC (36.5 ± 0.4 °C vs. 36.1 ± 0.3 °C *p* < 0.01). There was no effect of body fat on tympanic temperature (*p* = 0.44). There was no effect of age (*p* = 0.21) or body fat (*p* = 0.6) on weighted mean skin temperatures. There was no effect of age group (*p* = 0.17) or body fat (*p* = 0.22) on wellbeing scores.

## 4. Discussion

The main finding in this study was that whole body cryotherapy blunted the decrease in muscle torque following a downhill running bout that was observed in the control group, indicating that WBC may attenuate muscle damage and support post-exercise recovery. Young participants responded significantly better to WBC with regards to muscle torque retention when compared to the older participants. These results partially support the initial hypotheses, although there was little impact on the response to the downhill run and cryotherapy between participants of different body fat contents.

The 30-min downhill run caused a significant decrease in muscle torque for both cryotherapy and control participants, which is consistent with previous downhill running studies [23,24]. The average torque decrements were 6.4% and 11.5% for WBC and control, respectively. The decrease for the cryotherapy group is less severe than typically seen in other downhill running studies and this moderation effect could be significant in a sports and performance context.

To the authors’ knowledge, this is the first study to demonstrate a positive effect of a single treatment of WBC on muscle performance 24 h post-EIMD. Previous studies that have observed beneficial effects of WBC for treating EIMD either used multiple treatments [6,10] or partial body cryotherapy [9], where the head is not exposed to extreme cold, therefore having different physiological mechanisms [37]. Experiencing beneficial effects using just a single treatment of WBC highlights the potential effectiveness of the intervention and is likely to be more economical and feasible than applying multiple treatments. Caution should be exerted when interpreting the findings, since the overall interaction effect for group and time was non-significant (*p* = 0.10). Nonetheless, with a reasonably strong effect size of 0.67 for difference between groups in torque reductions, it is likely that such alleviations of muscle strength decrement following eccentrically biased exercise would result in superior athletic recovery.

The other EIMD markers however do not indicate support for the application of WBC post-downhill run. There was no difference in muscle soreness between the cryotherapy and control groups. The debate on whether WBC is effective in reducing muscle soreness has been highlighted previously [38,39] and by the discrepant findings between studies demonstrating benefits [6,13] versus studies that have not [28].

There was no effect of WBC on the blood CK response post-downhill run. CK is a commonly used EIMD marker due to its ease of detection in the circulation and the indication of disrupted muscle membranes [40]. The effects of WBC on CK levels are equivocal. The few studies that have demonstrated blunted CK responses [5,41] utilised multiple WBC treatments which might induce more attenuation of CK levels. The lack of impact of WBC on plasma CK post-downhill run indicates significant muscle fibre disruption. Associated characteristics include disrupted sarcomeres and Ca+ homeostasis, metabolic disturbances, Z-line streaming, presence of inflammatory markers and undermined excitation-contraction coupling [42,43], effects consistent with the reduction in muscle strength that was still observed in the WBC group. It is, therefore, evident that many of the physiological effects of muscle damaging exercise were present in the WBC participants for this study. WBC was also ineffective in enhancing mental wellbeing scores despite other studies indicating otherwise [44,45].

The reductions observed in tympanic and skin temperatures post-WBC are comparable to previous studies [20,46,47] and consistent with the notion that WBC causes a pronounced vasoconstriction response, ensuring that blood flow is diverted away from the extremities to protect internal organs. It is not clear to what extent this thermoregulatory response can support recovery and performance post-exercise, especially since it is unlikely that the skin temperature decreased low enough to illicit a significant analgesic response. Assessing same day performance measures (e.g., power tests) post-WBC may provide further understanding of how physiology responses post-cryotherapy might be linked to functional performance.

### 4.1. Effects of Age and Body Fat Content

To the authors’ knowledge, this is the first study to have investigated the effects of different ages and body fat contents on the response to WBC treatment for post-exercise recovery. Due to physiological differences between different age groups and body fat contents, it was hypothesised that younger and/or leaner men would respond more optimally to WBC post-exercise than older and/or higher fat individuals, respectively. The main finding of interest was that the young WBC participants’ muscle torques did not decrease 24 h after the downhill run, whereas it decreased substantially for the older group.

Despite this significant finding, the young WBC participants still experienced EIMD, since significant muscle soreness and elevated CK were observed. Nonetheless, muscle torque is considered the most important marker of muscle damage [48]. The more favourable response to WBC post-exercise observed in the younger participants (<40 years) may have implications for coaching and training programmes with the potential use of WBC to support recovery following eccentric muscle contractions. Half of the cryotherapy sample (13 of 26 participants) were aged below 40 and this sub-group did not experience muscle torque decrements to the extent of the participants aged 45 and above. Thus, WBC appeared to be particularly beneficial for the younger participants in a functional sense.

It is not clear why the younger participants would retain their muscle strength following the cryotherapy treatment more than the older participants. Owing to the established effects of ageing, potential theories include enhanced blood vessel and flow response to the leg muscles, better motor unit/muscle fibre activation, less disruption of excitation-contraction coupling, higher muscle-tendon stiffness, higher testosterone, reduced inflammation and/or the placebo effect. Due to the common occurrence of sarcopenia in elderly individuals [14], it is conceivable that the discrepancy in muscle recovery potential between age groups post-WBC and exercise can be attributed to differences in muscle mass. Enhanced muscle cooling is unlikely owing to the unusual finding of lower tympanic temperatures in the older participants post-WBC. Such theories and aspects would be potential revenues for further research to help understand how age differences can impact response to WBC post-exercise.

The leaner participants maintained their muscle torque post-WBC more than those with higher body fat, indicating that body fat could detriment the damage and recovery response to WBC post-exercise. Caution should be exercised in concluding this, since the interaction effect of body fat group was non-significant. Whilst it has been suggested that higher body fat decreases heat loss during cold exposure [15], there were no differences between HFAT and LFAT in tympanic and skin temperatures, which contrasts findings from previous WBC studies [20,21]. A possible explanation for this discrepancy is that tympanic and skin temperatures were only assessed at one time point post-WBC in this study. There was also no influence of body fat content on any of the other variables, contradicting the initial hypothesis. Further studies looking at the influence of different body compositions on response to WBC following other exercise bouts (e.g., sports fixtures, repeated sprints) might add more perspective on WBC applications for sports recovery.

### 4.2. Potential Limitations

The sample included a mixture of athletes from different sporting backgrounds and individuals who exercise recreationally. Most participants had a relative VO_2_ max below 50 mL/min/kg, of which a substantial portion were aged below 40. It can, therefore, be assumed that a large proportion of the sample were not trained athletes. It was initially the intention to include a variety of fitness levels and body sizes, but this factor should be considered before applying the findings to higher level sports practice.

A possible limitation is that muscle damage markers (strength, soreness and CK) were not assessed beyond 24 h post-exercise. Whilst this is the first study to examine the response to WBC following a downhill run, two previous studies have applied cold water immersions post-downhill run [49,50]. Both observed peak muscle damage markers 24 h post-run instead of 48 h. Other downhill run studies have observed peak muscle soreness at 24 h [37,51,52] and 48 h [53,54] without any meaningful difference between these first two days. The greatest inflammation and loss of muscle function also occurs within 24 h [55] and blood CK typically peaks at 24 h [48,56,57]. Additionally, the extent of muscle strength reduction indicates mild damage (only a 11.5% decrease for control group) where torque typically recovers within 48 h [58]. It is therefore conceivable that the damage response at 24 h would be a reliable indicator of damage extent at 48 h and any alleviation of EIMD at 24 h would likely result in quicker recovery to baseline. Athletes who train/compete several days a week are usually more concerned about next day recovery to successfully engage in further training sessions.

Finally, the logistical challenge of transporting participants to the cryotherapy chamber could have impacted some variables (e.g., tympanic temperatures) so it was not possible to control all ambient conditions before and after WBC.

## 5. Conclusions

Overall, WBC may alleviate the muscle damaging effects following downhill running due to an attenuation of muscle torque decreases. Despite EIMD being present, younger participants could take advantage of using WBC to mitigate muscle torque losses following an eccentrically biased long duration exercise. Body fat does not appear to heavily influence responses to WBC post-downhill run; however leaner individuals may benefit more by retaining levels of muscle strength. Future research should focus on the mechanisms through which younger practitioners can benefit more from cryotherapy treatments following an EIMD bout and how this could support recovery and sports performance. Additionally, it would be useful to further explore the potential recovery benefits of single WBC treatments following other exercise protocols.

## Figures and Tables

**Figure 1 ijerph-18-02906-f001:**
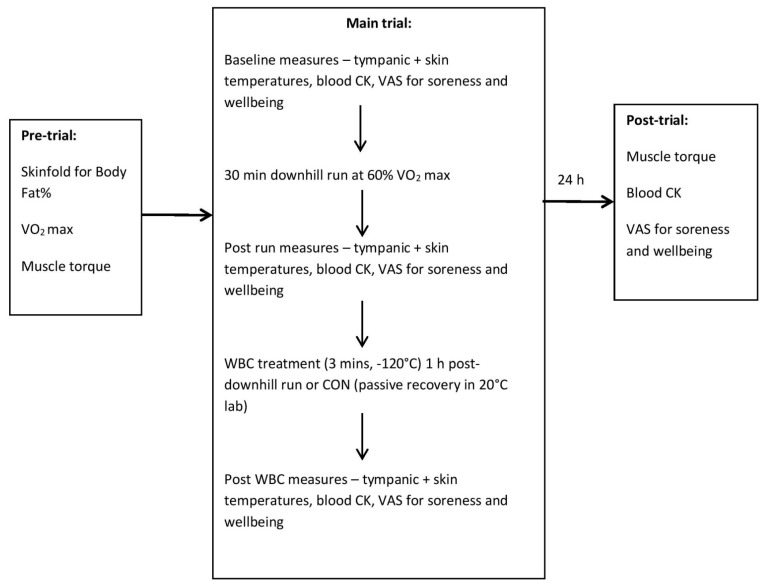
Protocol summary of measures for each trial. WBC—Whole Body Cryotherapy; CON—Control; CK—creatine kinase; VAS—visual analogue scale.

**Figure 2 ijerph-18-02906-f002:**
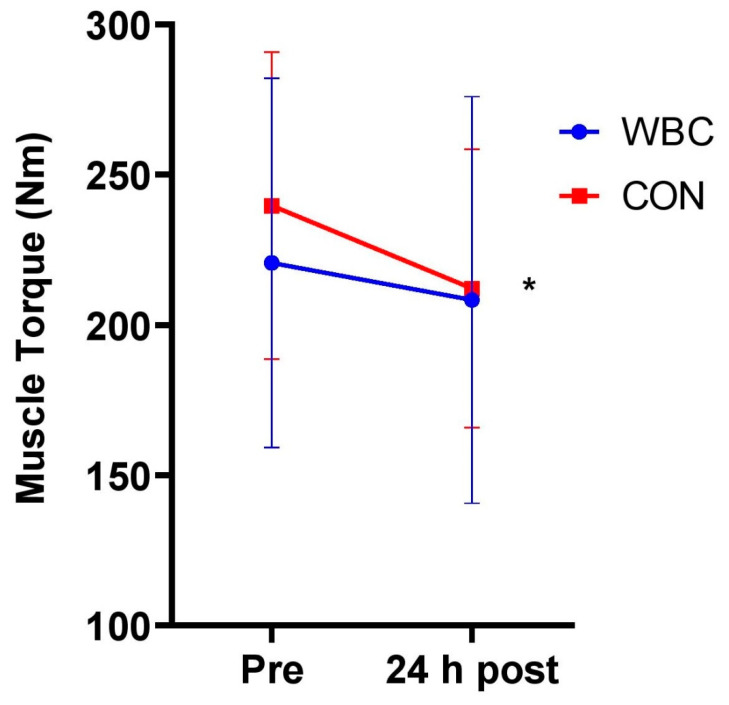
Maximal muscle torque response between WBC (n = 26) and CON (n = 15) groups. * *p* < 0.05 for decrease in both groups. Data presented as means ± standard deviations.

**Figure 3 ijerph-18-02906-f003:**
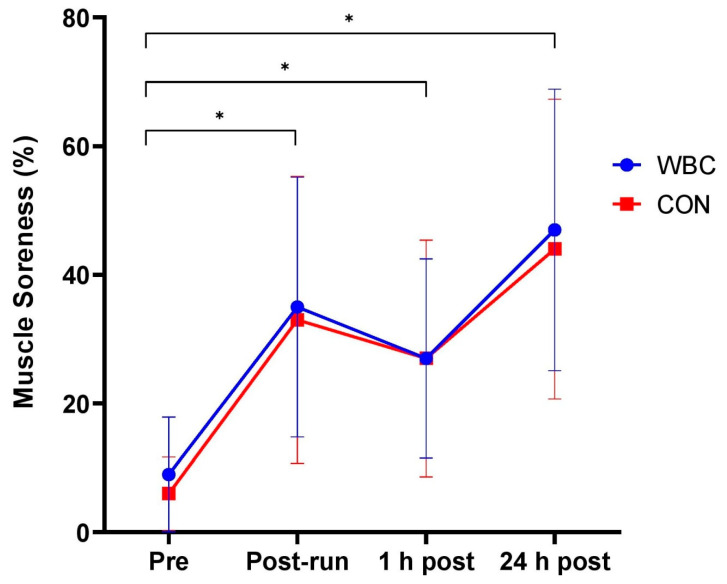
Muscle soreness response for WBC (n = 26) and CON (n = 15) groups. * *p* < 0.01 for increases from baseline for both groups. Data presented as means ± standard deviations.

**Figure 4 ijerph-18-02906-f004:**
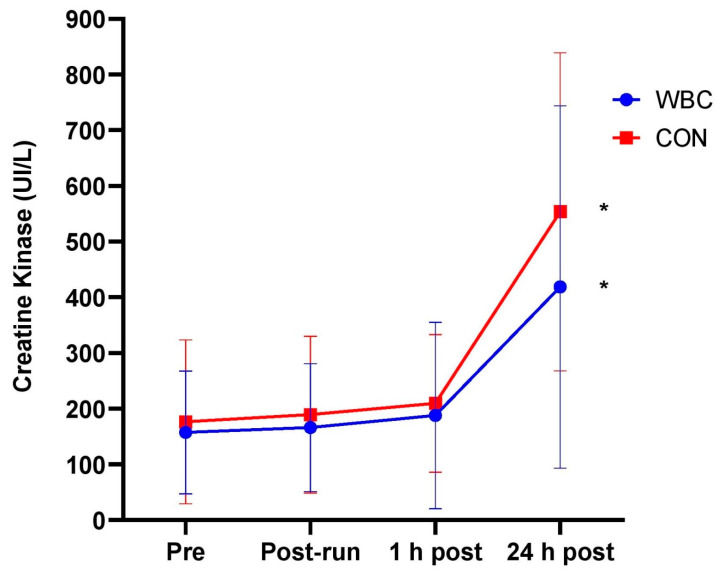
Blood CK response for WBC (n = 26) and CON (n = 15) groups. * *p* < 0.05 for increase from baseline in both groups. Data presented as means ± standard deviations..

**Figure 5 ijerph-18-02906-f005:**
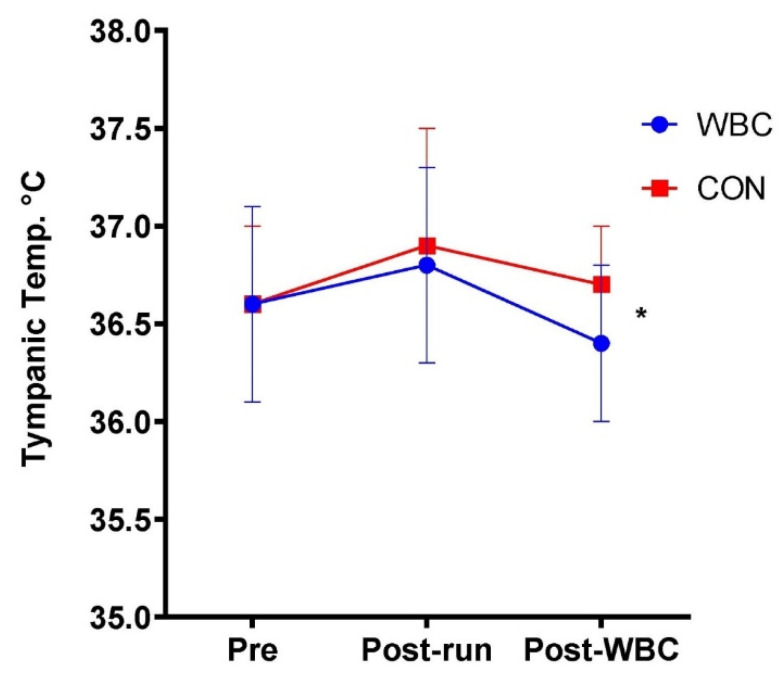
Tympanic temperature response for WBC (n = 26) and CON (n = 15) groups. * *p* < 0.01 for difference between groups at post-WBC. Data presented as means ± standard deviations.

**Figure 6 ijerph-18-02906-f006:**
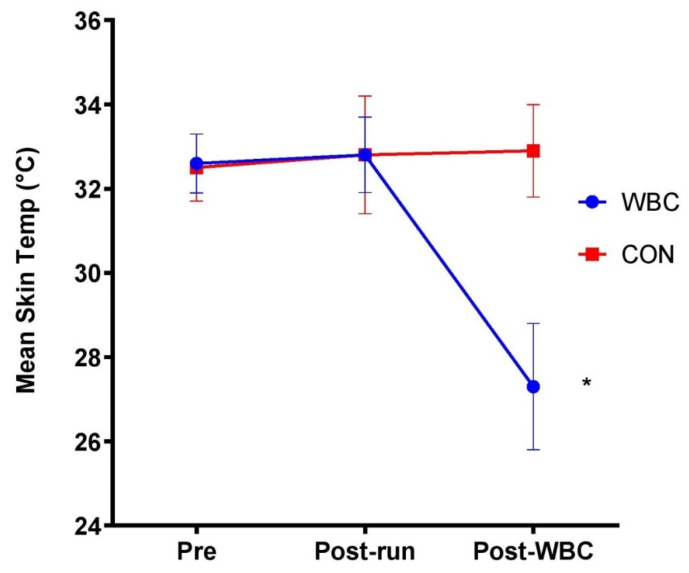
Skin temperature response for WBC (n = 26) and CON (n = 15) groups. * *p* < 0.01 for decrease in WBC group. Data presented as means ± standard deviations.

**Figure 7 ijerph-18-02906-f007:**
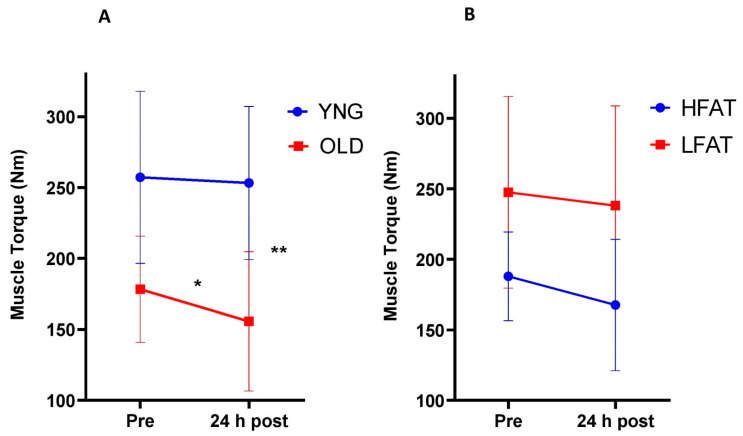
Maximal muscle torque response between YNG (<40 years, n = 13) and OLD (≥45 years, n = 10) participants (**A**), as well as between HFAT (≥20%, n = 10) and LFAT (≤15%, n = 8) participants (**B**) within WBC group. * *p* < 0.05 for decrease in OLD; ** *p* < 0.05 for interaction between age group and time. Data presented as means ± standard deviations.

**Table 1 ijerph-18-02906-t001:** Summary of characteristics for whole body cryotherapy (WBC) and control (CON) participants. Data presented as mean ± SD.

	WBC (n = 26)	CON (n = 15)	OVERALL (n = 41)	T Test between WBC and CON
Age (yrs)	41.8 ± 15.5	42.3 ± 10.4	42.0 ± 13.7	*p* = 0.93
Height (m)	1.78 ± 0.09	1.75 ± 0.06	1.76 ± 0.08	*p* = 0.21
Body mass (kg)	74.9 ± 10.8	75.6 ± 11.1	75.2 ± 10.8	*p* = 0.85
Body mass index (kg/m^2^)	23.7 ± 2.2	24.7 ± 2.9	24.1 ± 2.5	*p* = 0.22
Body fat %	18.8 ± 4.3	20.0 ± 4.9	19.2 ± 4.5	*p* = 0.4
Absolute VO_2_ max (L/min)	3.61 ± 0.55	3.53 ± 0.63	3.58 ± 0.57	*p* = 0.67
Relative VO_2_ max (mL/min/kg)	48.4 ± 5.1	46.8 ± 6.5	47.8 ± 5.6	*p* = 0.32

**Table 2 ijerph-18-02906-t002:** Wellbeing scores for WBC (n = 26) and CON (n = 15) groups. Data presented as means ± standard deviations.

	Pre	Post-Run	1 h Post	24 h Post	ANOVA Time Effect
WBC	83.8% ± 16.7	82.3% ± 16.0	85.5% ± 15.0	81.1% ± 20.2	*p* = 0.06
CON	78.9% ± 21.6	80.8% ± 18.4	81.2% ± 19.2	78.1% ± 16.9	*p* = 0.44

**Table 3 ijerph-18-02906-t003:** Results for all variables other than muscle torque between OLD (≥45 years, n = 10) and YNG (<40 years, n = 13) WBC participants.

Variable	WBC Age Group	Pre	Post-Run	1 h Post (Post-WBC)	24 h Post	*p* Value for Time × Group Interaction
Muscle Soreness	OLD	8.7% ± 7.2	28.6% ± 16.6	21.6% ± 15.8	42.6% ± 21.4	0.68
	YNG	10.1% ± 10.7	40.2% ± 22.7	31.3% ± 14.1	49.5% ± 21.8	
CK (UI/L)	OLD	149.5 ± 106.2	142.6 ± 94.5	175.9 ± 119.7	351.0 ± 283.3	0.22
	YNG	177.0 ± 111.9	196.1 ± 124.7	212.7 ± 194.2	502.1 ± 349.1	
Tympanic Temp	OLD	36.4 °C ± 0.5	36.7 °C ± 0.6	36.1 °C ± 0.3 *	n/a	0.17
	YNG	36.8 °C ± 0.4	36.8 °C ± 0.5	36.5 °C ± 0.4 *	n/a	
Skin Temp	OLD	32.6 °C ± 0.6	32.2 °C ± 0.7	27.4 °C ± 1.7	n/a	0.21
	YNG	32.8 °C ± 0.6	33.3 °C ± 0.7	27.4 °C ± 1.4	n/a	
VAS Wellbeing	OLD	88.5% ± 16.1	84.8% ± 13.3	87.7% ± 13.4	82.6% ± 21.6	0.17
	YNG	85.7% ± 10.5	84.8% ± 11.8	88.9% ± 8.2	85.1% ± 9.5	

* *p* < 0.01 for difference in tympanic temperature between groups post-WBC. Data presented as means ± standard deviations.

**Table 4 ijerph-18-02906-t004:** Results for all variables other than muscle torque between HFAT (≥20%, n = 10) and LFAT (≤15%, n = 8) WBC participants.

Variable	WBC Body Fat Group	Pre	Post-Run	1 h Post (Post-WBC)	24 h Post	*p* Value for Time × Group Interaction
Muscle Soreness	HFAT	9.1% ± 6.8	31.9% ± 16.7	24.8% ± 17.0	44.0% ± 22.0	0.78
	LFAT	12.25% ± 12.4	41.9% ± 20.8	34.1% ± 14.3	55.8% ± 17.2	
CK (UI/L)	HFAT	153.7 ± 103.3	147.1 ± 93.2	181.1 ± 118.3	391.2 ± 254.4	0.59
	LFAT	205.6 ± 131.4	233.7 ± 141.9	249.1 ± 234.2	588.3 ± 413.4	
Tympanic Temp	HFAT	36.5 °C ± 0.4	36.6 °C ± 0.6	36.2 °C ± 0.4	n/a	0.44
	LFAT	36.8 °C ± 0.4	36.7 °C ± 0.5	36.5 °C ± 0.4	n/a	
Skin Temp	HFAT	32.7 °C ± 0.7	32.4 °C ± 1.0	27.1 °C ± 2.0	n/a	0.6
	LFAT	32.9 °C ± 0.6	33.1 °C ± 0.7	27.3 °C ± 0.9	n/a	
VAS Wellbeing	HFAT	82.2% ± 22.8	80.7% ± 21.6	81.7% ± 21.9	79.9% ± 25.9	0.22
	LFAT	89.5% ± 9.4	80.8% ± 14.3	89.3% ± 9.6	80.1% ± 20.9	

Data presented as means ± standard deviations.

## Data Availability

Datasets are available on Pure Northampton. https://doi.org/10.24339/123cd0be-44c3-4638-9a06-33337afe07bb (accessed on 12 March 2021).

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
