# Peer review of "The Effects of Age and Body Fat Content on Post-Downhill Run Recovery Following Whole Body Cryotherapy"

_ijerph, 2021, doi:10.3390/ijerph18062906_

Round 1

Reviewer 1 Report

This is an interesting study that could provide meaningful information around mechanisms related to WBC's impact on muscle function.  The introduction was organized and appropriately set up the purpose.  The methods were clear and easy to follow.  The statistical approach and subsequently the results and interpretation need to address some gaps in how the data was analyzed.

1.  Table 1 presents data to demonstrate that there were not group differences at baseline in many of the factors.  However, muscle strength, the primary measure of the study, was not included in this analysis.  Figure 1 would suggest that determining if there are or are not differences in the pre muscle torque values is important.  Please report this t-test outcome.  The result of this t-test is critical to the subsequent data management.  If the groups are different at baseline, then subsequent analyses must account for this in some way.

2.  What was the rationale for only examining age and body fat within the WBC group?  What was the criteria for determining age (45 yrs does not really seem "old") and high/low fat cut points?  It all seems arbitrary, which reduces the value of interpretation.  Was there a statistical process used to create groups, perhaps creating even numbered groups?  Why are the CON and WBC groups different sizes to begin with?  Knowing if age and body fat also played a role or not in the CON group would strengthen the paper and be more aligned with a statistically sound approach.

3.  Given the study is a between groups design, instead of reducing the sample size even more by breaking into groups, did the authors consider an ANCOVA design where age and body fat are controlled for?  This would allow for a more statistically appropriate way to examine this influence between groups and avoid the arbitrary sub group creation.

4.  Creatine kinase was measured at 4 time points (pre, post, 1 hr post, and 24 hrs post) and data presented as such in figure 4.  The methods and interpretation of "baseline to 24 hrs post" seems to suggest the ANOVA was not a 2 (group) by 4 (time) ANOVA and instead just a 2x2.  If that observation is correct, then the methods and results need to be edited to indicate it was a 2x4 ANOVA.  As a general suggestion, it would be helpful for the reader to know how many ANOVA's were conducted and which were 2x2 and 2x4, etc.

5.  Table 3 reports p=0.17 with a reference to the  indicating p<0.01????  Not sure the * is needed by the interaction term if the intention is highlighting group differences earlier in the table.  It all reads very confusing.

6.  Figure 7:  were the pre muscle torque values different between groups?  This must be accounted for.

7.  The discussion may need to be edited based on adjustment for some of the prior statistical questions.  For example, no doubt the CON group had a greater decrease in maximal strength at 24 hours post (figure 1), but the CON group still ended at a higher torque level than the decreased torque in the WBC group.  So, what is the practical meaning of this then?  Further, unless you control for the greater starting torque, if may be that in the WBC group, the lack of change was simply related to the lower starting torque.  Conclusively stating this via statistical tests will only strengthen the interpretation of the age-related outcomes.  Until this is reported, it is difficult to conclude the WBC group responded differently.  

Reviewer 2 Report

The authors investigated the effects of age and body fat content on responses to Whole Body Cryotherapy following a downhill running bout. Results showed that Whole Body Cryotherapy may alleviate the muscle damaging effects following downhill running and younger participants could take advantage of using Whole Body Cryotherapy; also, leaner individuals may benefit more by retaining levels of muscle strength.

The topic is very interesting. The article is well written and methodologically correct. The introduction is clear and the methods are accurately described. Statistical techniques are appropriate and the results are described precisely and accurately. The discussions are relevant. I have only a few minor recommendations to make for improving the manuscript.

Line 10: Replace "41" with "Forty-one"

Lines 26-27: Remove the keywords already present in the title: "Whole Body Cryotherapy", "downhill running", "recovery", "age" and "body fat". If possible, replace them with different keywords. This will allow you to optimize the search for the manuscript through search engines.

Reviewer 3 Report

Major comments

The loss of muscle torque was less in the cryotherapy group, but they are quite different at baseline (220.6 cryo vs. 239.7 in control). Are these significantly different at baseline? What are the statistics on this? Given the decrease in force was greater in the control group, could it not be that the stronger group at baseline simply lost more? This would be supported by no difference in CK

What were the past/current exercise status of all groups? Could it be that the older group was relatively exercise naïve and thus saw a significant loss of torque whereas younger subjects were better trained and more resilient to EIMD?

In my humble opinion, the figures are not of publication quality. I would recommend remaking in software such as GraphPad Prism (available as a 30-day trial) as opposed to Excel.

Minor comments

Interesting result that there’s a loss of torque in old but not young.

Intro – excellent introduction!

103 – worth making this clear that (presumably) written informed consent was obtained before any screening or experimental procedures took place. I think you also mention later that this conformed to the Declaration of Helsinki – worth stating that here.

108-128 – where were the torque assessments completed, on an isokinetic dynamometer? Please give details of this.

143 – is there a reference for VAS? Same as the sentence after?

Line 192-194 – I think the way this is written is a little misleading in parts. I presume participants were asked if they had a medical history of cardiovascular disease or cancer and a negative answer was deemed ok to include in the study. As written, it sounds like you’ve done blood panels for cancer and screening for CV disease, which I presume is not the case?

Line 203-204 – good to see data handled properly when non-parametric. Is there evidence that these data sets are normally non-parametric in the literature or do you think it relates to your subject sample i.e. why do you think these data are non-parametric?

Line 203-204 – please give details of transformation e.g. were all data log10 transformed? Are data in figures the original data or transformed data? Worth adding a sentence to explain which data you are using.

Line 203-204 – is this not a two-way mixed model ANOVA as you have both a group variable i.e. old vs. young and time variable pre vs post?

Line 216 – please report p values to 3dp or as p <0.01

Line 219 – keep consistency in p values please – here reported only to 1dp

Figure 2 – what is the statistics on WBC vs. CON at baseline? They look like they are quite different before the intervention. Is it possible to closer match the groups at baseline (using your existing data) and see if the WBC effect still exists?

Figure 3 – please report n numbers per group as opposed to the n=41 overall

Figures 3 and 4 look different – suggest keeping markers and line colours consistent throughout. I would also suggest using publication quality software to create your figures such as GraphPad Prism as opposed to Excel.

Figure 4 – remove the decimal points on y axis and report the n number pre group as opposed to the n overall

Figure 5 – use symbol for degrees rather than write deg. Include n numbers per group

Figure 6 – expand the y axis – no need to show data except ~24-34 degrees. Report n numbers per group

263 – Can you add a statistics column to this table please so we can see what was approaching significance?

Figure 7 – remove decimal points on y axis

301 – given groups were not particularly well matched at baseline I would advise caution in reporting this outcome; it may simply be that those who were stronger to begin with lost more force with EIMD. Could you do some additional analysis to look at the % loss of force for each individual participant i.e. express loss of force relative to baseline – this would partially address this point

321 – report p value to 2dp

335-337 –disruption to the sarcomeres, Ca2+ homeostasis etc. also causes a disruption to energy metabolism, which may explain some of the whole-body observations in reduced exercise performance (Etheridge et al. 2014, FASEB 29 (4), 1235-1246)

358 – are there data to report on training status of the older and younger groups? Were they matched for activity levels? If not, it may be that the older individuals were exercise naïve and thus more susceptible to eccentric damage, which has been well documented in the literature. Might be useful to add a couple of sentences to discuss this aspect

370 – 380 – could it also be that the older subjects are more inflamed at baseline and thus an increase in inflammation with eccentric damage? So-called inflammaging?

Informed consent statement – was this written informed consent – if so, please state this

Reviewer 4 Report

This study investigates the effects of age and body fat content on post-downhill run 2
recovery following whole body cryotherapy. The research question is original and useful. However, I have some revision to suggest.

Introduction: The introduction needs to be clear what the practical question is that you are trying to address. How the answer to this question is important to the field as this is not clear or obvious? How is this study and impactful study and not trivial as this needs more clarity as well. The key issue here is to make sure you set up your approach to the problem. The approach to the problem is essential in determining and describing the rationale for the study. You have not given a basic rationale for the choices made for the variables used in the study. Please treat to improve this part of the Introduction section

Methods: Why did not you perform a power analysis to determine the sample size? If it has been done, please report the results

Discussion: The novelty of the present study is largely diluted by findings that simply tend to confirm previous published evidence making the study descriptive and giving it a rather low priority. It is recommended to elaborate and emphasize some potentially novel aspects presented here. I suggest considering the following studies about recovery and/or cryotherapy:

- Recovery Time Profiling After Short-, Middle- and Long-Distance Swimming Performance. J Strength Cond Res. 2019 May;33(5):1408-1415. doi: 10.1519/JSC.0000000000002066.
- Physiological responses to partial-body cryotherapy performed during a concurrent strength and endurance session. Appl Physiol Nutr Metab. 2019 Jan;44(1):59-65. doi: 10.1139/apnm-2018-0202.

Round 2

Reviewer 1 Report

Thank you for addressing the review comments.

Reviewer 3 Report

Dear authors,

Thank you for taking my feedback on board and thank you for your rebuttal. You provide good explanations to points I have queried and the manuscript has been changed where required. I now recommend this for publication.